# Factors Associated with Low Albumin in Community-Dwelling Older Adults Aged 75 Years and Above

**DOI:** 10.3390/ijerph20216994

**Published:** 2023-10-29

**Authors:** Kazunari Kobayashi, Tomoko Nishida, Hisataka Sakakibara

**Affiliations:** 1Department of Nursing, Graduate School of Medicine, Nagoya University, Nagoya 461-8673, Japan; 2Department of Nursing, Graduate School of Medicine, Gifu University, Gifu 501-1194, Japan; nishida.tomoko.t0@f.gifu-u.ac.jp; 3School of Nursing, Ichinomiya Kenshin College, Ichinomiya 491-0063, Japan; h.sakakibara.t@ikc.ac.jp; 4Graduate School of Medicine, Nagoya University, Nagoya 461-8673, Japan

**Keywords:** albumin, physical activity, oral function, weight loss

## Abstract

This study aimed to clarify the factors associated with low albumin in Japanese older adults aged ≥75 years. The data utilized were the health checkup data of older adults aged ≥75 years from 42 municipalities in Gifu Prefecture, which were provided by the National Health Insurance database system. After excluding the data of individuals with incomplete information on serum albumin, BMI, lifestyle habits, or weight at previous year, the data from 18,674 individuals’ health checkup were analyzed. A logistic regression showed that low albumin was associated with smoking, not walking at least 1 h/day, slow walking speed, difficulty in chewing, slow eating speed, weight loss in the previous year, and underweight. Furthermore, an analysis conducted for older adults aged ≥85 years showed that low albumin was associated with not walking at least 1 h/day, difficulty in chewing, slow eating speed, weight loss in the previous year, and underweight. In the future, the number of older adults will increase in Japan; therefore, a strategic approach to promote the health of these aged individuals will become even more necessary. An early approach to maintaining an active physical life, an appropriate weight, and good oral function will lead to improved health in older adults.

## 1. Introduction

Albumin is the major plasma protein produced primarily in the liver and is the most abundant protein in the extracellular compartment [1]. Serum albumin has been used as a primary nutritional assessment marker. Recently, it has also been reported that albumin is a negative acute phase protein, whose levels are lowered by inflammation in the body, such as chronic diseases [2]. This could be due to an underlying inflammatory response [3,4].

Low albumin levels pose a health risk and are effective markers for the early detection of health risk in older adults. Low albumin is reportedly associated with increased incidence of all-cause mortality, cancer-related mortality, cardiovascular disease, respiratory disease, and other adverse outcomes [5,6]. This association was also evident in a meta-analysis of 48 cohort study articles [6]. Similar results were observed in studies of older adults [7,8,9]. A large-scale survey of 70,000 community-dwelling older adults aged ≥65 years demonstrated a graded inverse relationship between lower serum albumin levels and increased risk of adverse health effects including death [8]. The study reported that the risk of all-cause mortality and cardiovascular mortality was two times greater in older adults with albumin levels <3.8 g/dL than in those with levels ≥4.4 g/dL. Additionally, the mortality risk from respiratory disease was double in patients with albumin levels <4.0 g/dL. In older adults, even a mild reduction in the 3.5 g/dL threshold of albumin used in clinical practice can pose a risk.

Using low albumin levels to explore the factors associated with health risk in older adults is beneficial from a primary prevention perspective. Many studies examining low albumin levels in older adults have focused on frailty and sarcopenia [10,11,12,13,14,15,16]. These studies analyzed the association between serum albumin and muscle mass, walking speed, grip strength, and other parameters. A study in Taiwan of community-dwelling older adults aged 65–85 years reported that grip strength was positively associated with serum albumin in both men and women [12]. Additionally, the estimated appendicular muscle mass was positively associated with serum albumin in men. A study by Schalk et al. [13] examined the cross-sectional and longitudinal relationship between serum albumin and muscle strength in older adults aged 65–88 years, and showed that low albumin levels are associated with current low muscle strength as well as declining muscle strength in the future.

Meanwhile, some studies reported no significant relationship between muscle mass and albumin in older adults [14], or showed a statistically significant but negligible relationship [15]. In a survey involving 172 healthy young people (aged 18–30 years) and 271 older adults (aged 69–81 years) from the European MYOAGE study, serum albumin correlated positively with lean mass percentage in the young individuals but not in the older adults. However, no association was reported between serum albumin and grip strength and walking speed in both young and older adults [16].

In Japan, health checkups are conducted by the local governments for older adults aged ≥75 years to ensure early detection of chronic diseases and early detection and early support of frailty associated with aging. Among the health checkup parameters, serum albumin is measured by many local governments to screen for health risks.

In this study, the data of 18,674 health checkups conducted in 42 municipalities in Gifu Prefecture from April 2019 to March 2020 were used to clarify the factors associated with low albumin levels in older adults aged ≥75 years. Particularly, we focused on parameters such as frailty-related physical activity, oral function, and weight loss, and examined the association in older adults aged ≥75 years as well as oldest-old adults aged ≥85 years.

## 2. Material and Methods

### 2.1. Participants

In this study, we analyzed the health checkup data of older adults aged ≥75 years from 42 municipalities in Gifu Prefecture. The data utilized were from health checkups conducted between April 2019 and March 2020, which were provided by the National Health Insurance database system (Kokuho Database) managed by the Gifu Prefecture Federation of National Health Insurance Organizations. The Kokuho Database includes the data from health checkups, medical, and nursing care, from the National Health Insurance for individuals aged <75 years and the Latter-stage Elderly Medical Care System for individuals aged ≥75 years among residents in each prefecture. The health checkup for the elderly aged 75 and over is conducted under the jurisdiction of the Ministry of Health, Labour, and Welfare using a standardized protocol, and the data are registered in this system.

As of May 2021, when the data were extracted from the system, there were 327,498 system registrants aged ≥75 years. Among them, 70,189 had health checkup data from April 2019 to March 2020. There were 27,303 individuals with no missing data for serum albumin levels, height, weight, and lifestyle habits. Since we had to calculate the weight change from the previous year, 8629 individuals without weight data from the checkup in the previous year were excluded. The remaining 18,674 individuals’ health checkup data were analyzed. The health checkups were conducted once a year, and the data analyzed in this study did not include multiple sets of health checkup data of the same individual.

This study was conducted after receiving approval from the bioethics review committee at Nagoya University Graduate School of Medicine (Approval number: 17-115-2).

### 2.2. Survey Content

We used data on age, sex, body composition, blood data, lifestyle habits, and treatment status. In Japan, the health check-up for the elderly aged 75 and over is conducted under the jurisdiction of the Ministry of Health, Labour, and Welfare using a standardized protocol. Therefore, the physical measurement items, blood test items, and lifestyle questionnaires are nearly uniform in the health checkup of all 42 municipalities in Gifu Prefecture covered in this study.

#### 2.2.1. Body Composition

The height and weight measured during the health checkups were used. The body mass index (BMI) was determined as weight (kg) divided by height squared (m^2^). Based on the BMI values, the World Health Organization (WHO) body type classification and Global Leadership Initiative on Malnutrition (GLIM) criteria [17] were used to classify the participants into the following categories: underweight, <18.5 kg/m^2^; slightly underweight, ≥18.5 kg/m^2^ and <20 kg/m^2^; normal, ≥20 kg/m^2^ and <25 kg/m^2^; and overweight, ≥25 kg/m^2^.

#### 2.2.2. Weight Loss in the Previous Year

Weight loss was calculated as the rate of decrease in the weight from the previous year as follows: (weight at checkup [kg] − weight at previous year’s checkup [kg])/weight at previous year’s checkup (kg) × 100. Based on the weight loss in the previous year, the participants were categorized into the following categories: ≤5%, 5–10%, and >10% weight loss.

#### 2.2.3. Blood Data

The serum albumin (g/dL) data were collected. Previous studies on the general population have demonstrated that the inverse relationship between serum albumin and risk of death, hospitalization, and frailty are present [7,8,10]. A cut-off value of 3.8 g/dL on albumin has been used among community-dwelling older adults, and has been shown to have a high risk of sarcopenia and mortality risk [18,19,20]. In this study, the threshold for low serum albumin was set at 3.8 g/dL.

#### 2.2.4. Lifestyle Habits

Answers to the following questions were obtained from the health checkup questionnaires of the participants:“I walk or perform equivalent physical activity for at least 1 h/day in my daily life” (yes/no)“I walk faster than people of the same age and sex” (yes/no);Chewing situation (able to chew anything/difficulty in chewing);“I eat faster than other people” (fast/normal/slow);“I skip breakfast three or more times a week” (yes/no);Smoking habits (smoke/do not smoke);Drinking habits (drink daily/drink occasionally/rarely drink);“I get enough rest from sleep” (yes/no);

#### 2.2.5. Medical Treatment Status

The National Health Insurance database system that provided the data for this study records information on medical treatment. Based on information from medical institutions’ consultations from April 2019 to March 2020, we used data on whether there was a claim for the following diseases:Metabolic diseases (diabetes, dyslipidemia, hyperuricemia, or fatty liver);Musculoskeletal diseases (osteoporosis, joint diseases, spinal column disorders, fractures, or other musculoskeletal diseases);Cardiovascular diseases (hypertension or other cardiovascular diseases);Kidney diseases (diabetic nephropathy, chronic kidney failure, dialysis, or other kidney diseases);Respiratory inflammatory diseases (chronic obstructive pulmonary disease, infectious pneumonia, or aspiration pneumonia);Malignant neoplasms.Thus, we summarized the presence or absence of claim receipts for the aforementioned conditions and determined the current medical treatment status of each participant.

### 2.3. Statistical Analysis

The *t*-test or χ^2^ test was used to examine the associations between low albumin levels and sex, age, treatment status, lifestyle habits, body composition, and weight loss.

To determine the factors associated with low albumin, we first set the dependent variable as 0 (albumin level ≥ 3.8 g/dL) and 1 (albumin level < 3.8 g/dL). We then input the variables of lifestyle habits or weight loss in the previous year as independent variables and conducted a logistic regression analysis adjusting for sex and age. Following this, sex, age, metabolic disease, musculoskeletal disease, cardiovascular disease, kidney disease, respiratory inflammatory disease, and malignant neoplasm were input as moderator variables, and a multivariate logistic regression analysis was conducted with all variables related to lifestyle habits and body weight as independent variables. A similar analysis was conducted for older adults aged ≥85 years.

All statistical analyses were conducted using Statistical Product and Service Solutions 27.0 for Windows (IBM, Tokyo, Japan); *p*-values < 0.05 were considered statistically significant.

## 3. Results

Table 1 shows the characteristics of participants based on albumin levels. Among the 18,674 participants, 55.7% (*n* = 10,400) were female, and 44.3% (*n* = 8274) were male. The average age was 81.3 years with a standard deviation (SD) of 4.2 years. The prevalence of low albumin levels was 8.1% (*n* = 1514). Male sex (*p* < 0.001), older age (*p* < 0.001), low body mass index (*p* < 0.001), and weight loss in the previous year (*p* < 0.001) were significantly associated with low albumin. Low albumin levels were observed in 6.3% (*n* = 924) of older adults aged 75–84 years, 13.3% (*n* = 382) of those aged 85–89 years, and 20.2% (*n* = 208) of those aged ≥90 years (data not shown). Of the total participants, 61.7% had metabolic diseases, 73.8% had musculoskeletal diseases, 74.7% had cardiovascular diseases, 7.5% were kidney diseases, 28.5% had respiratory inflammatory diseases, and 11.6% had malignant neoplasms. Almost all diseases were significantly associated with low albumin.

Table 2 shows the association of low albumin with lifestyle habits and body weight. Low albumin was set as the dependent variable, variables related to lifestyle habits and body weight were set as the independent variables, and logistic regression analysis was conducted after adjusting for sex and age. The results showed that all variables except skipping breakfast were associated with low albumin levels. We then adjusted for sex, age, metabolic disease, musculoskeletal disease, cardiovascular disease, kidney disease, respiratory inflammatory disease, and malignant neoplasms, and included all lifestyle habits and body weight variables as independent variables. Regarding lifestyle habits, walking at least 1 h/day (odds ratio [OR]: 0.78, 95% confidence interval [CI]: 0.70–0.88) and fast walking speed (OR: 0.81, 95% CI: 0.72–0.91) showed significantly lower odds of low albumin levels. Difficulty in chewing (OR: 1.34, 95% CI: 1.19–1.51), slow eating speed (OR: 1.27, 95% CI: 1.10–1.46), and smoking (OR: 1.39, 95% CI: 1.10–1.77) showed significantly higher odds of low albumin levels. Those who drank occasionally (OR: 0.77, 95% CI: 0.66–0.91) showed lower odds of low albumin levels than those who drank rarely; however, no association was noted in those who drank daily. For associations with variables related to body weight, 5–10% weight loss in the previous year (OR: 1.45, 95% CI: 1.21–1.74), ≥10% weight loss in the previous year (OR: 2.95, 95% CI: 2.05–3.94), and underweight BMI classification (<18.5 kg/m^2^) (OR: 1.36, 95% CI: 1.14–1.61) showed significantly higher odds of low albumin levels.

Table 3 shows the results of a similar analysis conducted for those aged ≥85 years. Multivariate logistic regression analysis, adjusted for sex, age, metabolic disease, musculoskeletal disease, cardiovascular disease, kidney disease, respiratory inflammatory disease, and malignant neoplasm, was conducted with all variables related to lifestyle habits and body weight entered as independent variables. The results showed that walking for at least 1 h/day (OR: 0.69, 95% CI: 0.57–0.83) had significantly lower odds of low albumin levels, whereas difficulty in chewing (OR: 1.40, 95% CI: 1.15–1.69) and slow eating speed (OR: 1.26, 95% CI: 1.01–1.57) had significantly higher odds of low albumin levels. For associations with variables related to body weight, ≥10% weight loss in the previous year (OR: 2.74, 95% CI: 1.69–4.43) and underweight BMI classification (<18.5 kg/m^2^) (OR: 1.38, 95% CI: 1.06–1.80) showed significantly higher odds of low albumin levels.

## 4. Discussion

This study showed that even after adjusting for age, sex, and presence/absence of medical treatment and disease, low albumin levels (<3.8 g/dL) were associated with smoking, not walking at least 1 h/day, slow walking speed, difficulty in chewing, slow eating speed, weight loss in the previous year, and underweight BMI classification (<18.5 kg/m^2^). Furthermore, a similar analysis conducted for older adults aged ≥85 years showed that low albumin levels were associated with not walking at least 1 h/day, difficulty in chewing, slow eating speed, weight loss in the previous year, and underweight BMI classification (<18.5 kg/m^2^).

The WHO guidelines on physical activity and sedentary behavior recommend performing at least 300 min/week of moderate-intensity aerobic physical activity for older adults aged ≥65 years to improve their health [21]. In this study, although the activity intensity while walking for at least 1 h/day could vary depending on the older adult, the activity time was longer than that recommended in the WHO guidelines, and the older adults were effectively living an active life. Additionally, older adults who walk faster than their peers are expected to be quite active and maintain their muscle strength and physical function [22,23,24,25]. The results of this study show the association between high daily physical activity and low albumin levels.

To the best of our knowledge, only one study has examined the association between serum albumin levels and physical activity in community-dwelling residents, including older adults [26]. The study by Dupont et al. [26] investigated the association between serum albumin and physical activity in 2577 males aged 40–79 years (average age, 59.7 ± 11.0 years), and the univariate linear regression analysis results showed a significant positive association. However, the analysis conducted by adjusting for age, BMI, smoking, and other factors did not reveal a significant association. When the participants were divided into the 40–59 years and 60–79 years age groups, the older adults showed higher β-coefficient and 95% CI values than the younger adults; the former may have presented a positive association between serum albumin levels and physical activity. Another study examining the effect of exercise intervention in older adults (average age, 83 ± 3 years) in public or private residential care facilities showed that normal albumin levels were maintained in them [27].

Physical activity has a systemic anti-inflammatory effect [28]. A meta-analysis study of middle-aged and older post-menopausal women reported that physical activity significantly improved the levels of interleukin-6, tumor necrosis factor-α, and C-reactive protein inflammatory markers [29,30]. The serum albumin assessed as the outcome in this study is a negative acute-phase protein whose level decreases during inflammation [31], and this could be attributed to albumin levels being maintained by the anti-inflammatory effect of high levels of daily physical activity.

In older adults aged ≥75 years in this study, weight loss ≥5% from the previous year (5–10% weight loss, OR: 1.45; ≥10% weight loss, OR: 2.85) and underweight BMI classification (<18.5 kg/m^2^) were associated with low albumin levels. However, no association was reported between low albumin levels and overweight BMI classification (≥25 kg/m^2^), which is expected to be associated with the risk of chronic inflammation. Weight loss is used worldwide to diagnose malnutrition. In the GLIM criteria [17], weight loss ≥5% within the past 6 months or ≥10% over the past ≥6 months is listed as one of the diagnostic criteria for malnutrition. Although low BMI is defined differently depending on the community, the GLIM criteria [17] listed Asians aged ≥70 years with a BMI <20 kg/m^2^ as one of the diagnostic criteria. Weight loss in the previous year and underweight BMI classification in this study were most likely due to a state of malnutrition. Additionally, there was no association between low albumin and overweight BMI classification in this study; hence, our results could be due to the association between malnutrition and low albumin levels.

Furthermore, “difficulty in chewing” and “slow eating speed” were associated with low albumin levels, independent of BMI and weight loss in the previous year. These two are the main symptoms experienced by older adults with oral frailty [32] and are used in screening for dysphagia [33]. A study by Motokawa et al. [34] examined the association between masticatory ability and low albumin levels (<4.0 g/dL) in Japanese older adults (average age, 73.9 years) and reported that decreased masticatory strength was associated with low albumin, even after adjusting for BMI and energy intake, which corroborated our results.

Maintenance of oral function not only affects the quantity of food consumed, but also the content/type of food consumed [35]. The study by Motokawa et al. [34] also investigated the nutritional intake of older adults and found that the intake of all nutrients, except carbohydrates, reduced significantly in older adults with decreased masticatory power. Another study [36] investigated a general population of adults aged ≥40 years in Japan and found that decreased overall oral function, including masticatory ability, was associated with reduced intake of legumes and meat in both men and women. Moreover, it was significantly and independently associated with protein intake below the reference value based on the Japanese intake standards (OR: 1.70, 95% CI: 1.21–2.35). Similarly, among the participants in this study, older adults with difficulty in chewing and slow eating speed may have had poor oral function and malnutrition due to a biased diet.

The limitations of nutritional evaluation using albumin have been reported in recent years [2,3]. It has also been reported that patients’ serum albumin and pre-albumin levels do not decrease until severe starvation [4]. However, our results suggest that weight loss in the previous year, underweight BMI classification, and decreased oral function in older adults aged ≥75 years are associated with malnutrition and risk of low albumin levels.

The results of this study revealed that even among older adults aged ≥85 years, walking for at least 1 h/day, difficulty in chewing, slow eating speed, weight loss in the previous year, and underweight BMI classification (<18.5 kg/m^2^) were associated with low albumin levels. Older adults are more likely to have diseases or malnutrition, thus increasing the risk of low albumin levels [37]. In this study, low albumin levels <3.8 g/dL were observed in 6.3% of older adults aged 75–84 years, 13.3% of those aged 85–89 years, and 20.2% of those aged ≥90 years. In the future, the number of older adults aged ≥85 years will increase in Japan; therefore, a strategic approach to promote the health of these aged individuals will become even more necessary. An early approach to maintaining an active physical life, an appropriate weight, and good oral function at the age of ≥85 years will lead to improved health in the oldest-old adults.

This study has several limitations. This study adopted a cross-sectional design; therefore, it cannot show causality. We used the results of health checkups for older adults conducted by municipalities in Gifu Prefecture. The checkups were conducted at medical institutions or venues designated by the municipalities, and they do not include older adults who were hospitalized or those who had difficulties going to the checkup locations. In this study, among overall 327,498 older adults, there were 70,189 with health checkups, and the 27,303 with no missing data. Therefore, there is a possibility that the data collected was from relatively healthy individuals. Additionally, medical checkups are conducted only for those who wish to receive them; therefore, it is possible that the number of people interested in health or conscious of leading a healthy life is increasing.

Responses to the questionnaire in this study were self-reported and included subjective statements such as “I walk faster than people of the same age and sex”. A previous study has demonstrated that subjectively faster walking speeds are associated with a lower risk of heart failure and a variety of cardiovascular disease events in the general population, and has reported the utility of subjective walking speed in primary prevention [38]. However, it has limitations in terms of the accuracy of its responses.

## 5. Conclusions

This study found that low albumin was associated with smoking, not walking at least 1 h/day, slow walking speed, difficulty in chewing, slow eating speed, weight loss in the previous year, and underweight. Furthermore, a similar analysis conducted for older adults aged ≥85 years showed that low albumin levels were associated with not walking at least 1 h/day, difficulty in chewing, slow eating speed, weight loss in the previous year, and underweight. In the future, the number of older adults will increase in Japan; therefore, a strategic approach to promote the health of these aged individuals will become even more necessary. An early approach to maintaining an active physical life, an appropriate weight, and good oral function will lead to improved health in older adults included aged ≥85 years.

## Figures and Tables

**Table 1 ijerph-20-06994-t001:** The characteristics of participants based on albumin levels.

	Total	Normal (≥3.8 g/dL)	Low Albumin (<3.8 g/dL)	*p*
Age (years)	81.34 ± 4.24	81.14 ± 4.10	83.64 ± 5.02	<0.001
BMI (kg/m^2^)	22.36 ± 3.10	22.40 ± 3.07	21.94 ± 3.40	<0.001
Weight loss in the previous year (%)	−0.50 ± 3.67	−0.44 ± 3.53	−1.20 ± 4.90	<0.001
Sex				
Female	10,400 (55.7)	9652 (56.2)	748 (49.4)	<0.001
Male	8274 (44.3)	7508 (43.8)	766 (50.6)	
Metabolic diseases	11,517 (61.7)	10,619 (61.9)	898 (59.3)	0.052
Musculoskeletal diseases	13,789 (73.8)	12,582 (73.3)	1207 (79.7)	<0.001
Cardiovascular diseases	13,953 (74.7)	12,732 (74.2)	1221 (80.6)	<0.001
Kidney diseases	1398 (7.5)	1210 (7.1)	188 (12.4)	<0.001
Respiratory inflammatory diseases	5315 (28.5)	4771 (27.8)	544 (35.9)	<0.001
Malignant neoplasms	2159 (11.6)	1907 (11.1)	252 (16.6)	<0.001

Data are expressed as mean ± standard deviation (SD) or frequency (%). *p* values by *t*-test or χ^2^ test.

**Table 2 ijerph-20-06994-t002:** The association of low albumin with lifestyle habits and body weight.

	Normal (≥3.8 g/dL)	Low Albumin (<3.8 g/dL)	Adjusted Sex and Age			Multiple Adjustment ^†^		
	*n* (%)	*n* (%)	OR (95% CI)	*p*	Overall *p*	OR (95% CI)	*p*	Overall *p*
I walk or perform equivalent physical activity for at least 1 h/day in my daily life								
Yes	9541 (55.6)	656 (43.3)	0.67 (0.61–0.75)	<0.001		0.78 (0.70–0.88)	<0.001	
No	7619 (44.4)	858 (56.7)	reference			reference		
I walk faster than people of the same age and sex								
Yes	8306 (48.4)	547 (36.1)	0.67 (0.60–0.75)	<0.001		0.81 (0.72–0.91)	0.001	
No	8854 (51.6)	967 (63.9)	reference			reference		
Chewing situation								
Able to chew anything	12,871 (75.0)	987 (65.2)	reference			reference		
Difficulty in chewing	4289 (25.0)	527 (34.8)	1.50 (1.34–1.68)	<0.001		1.34 (1.19–1.51)	<0.001	
I eat faster than other people								
Fast	2941 (17.1)	213 (14.1)	0.96 (0.82–1.12)	0.604	<0.001	0.98 (0.84–1.15)	0.841	0.007
Normal	11,944 (69.6)	968 (63.9)	reference			reference		
Slow	2275 (13.3)	333 (22.0)	1.55 (1.35–1.78)	<0.001		1.27 (1.10–1.46)	0.001	
I skip breakfast three or more times a week								
Yes	546 (3.2)	65 (4.3)	reference			reference		
No	16,614 (96.8)	1449 (95.7)	0.85 (0.65–1.12)	0.249		0.99 (0.75–1.30)	0.933	
Smoking habits								
Smoke	16,408 (95.6)	1421 (93.9)	reference			reference		
Do not smoke	752 (4.4)	93 (6.1)	1.49 (1.18–1.87)	0.001		1.39 (1.10–1.77)	0.006	
Drinking habits								
Drink daily	3014 (17.6)	267 (17.6)	0.86 (0.74–1.01)	0.058	0.005	0.91 (0.77–1.07)	0.242	0.054
Drink occasionally	3141 (18.3)	205 (13.5)	0.71 (0.60–0.83)	<0.001		0.77 (0.66–0.91)	0.002	
Rarely drink	11,005 (64.1)	1042 (68.8)	reference			reference		
I get enough rest from sleep								
Yes	13,759 (80.2)	1203 (79.5)	reference			reference		
No	3401 (19.8)	311 (20.5)	1.15 (1.01–1.31)	0.039		0.99 (0.86–1.13)	0.863	
Weight loss in the previous year								
≤5%	15,916 (92.8)	1285 (84.9)	reference		<0.001	reference		<0.001
5–10%	1089 (6.3)	169 (11.2)	1.66 (1.39–1.98)	<0.001		1.45 (1.21–1.74)	<0.001	
>10%	155 (0.9)	60 (4.0)	3.92 (2.87–5.37)	<0.001		2.85 (2.05–3.94)	<0.001	
BMI categories								
Underweight	1576 (9.2)	220 (14.5)	1.63 (1.39–1.92)	<0.001	<0.001	1.36 (1.14–1.61)	<0.001	0.006
Slightly underweight	2098 (12.2)	203 (13.4)	1.15 (0.98–1.36)	0.092		1.09 (0.92–1.28)	0.328	
Normal	10,251 (59.7)	824 (54.4)	reference			reference		
Overweight	3235 (18.9)	267 (17.6)	1.06 (0.92–1.23)	0.404		1.04 (0.90–1.21)	0.618	

Data are expressed as odds ratios (OR) and 95% confidence intervals (CIs) upon multiple logistic regression analysis. ^†^ Adjusted for sex, age, metabolic diseases, musculoskeletal diseases, cardiovascular diseases, kidney diseases, respiratory inflammatory diseases, malignant neoplasms, walking at least 1 h/day, walking speed, chewing situation, eating speed, skipping breakfast, smoking habits, drinking habits, getting enough rest from sleep, weight loss in the previous year, BMI categories.

**Table 3 ijerph-20-06994-t003:** The association of low albumin with lifestyle habits and body weight among older adults aged ≥85 years.

	Normal (≥3.8 g/dL)	Low Albumin (<3.8 g/dL)	Adjusted Sex and Age			Multiple Adjustment ^†^		
	*n* (%)	*n* (%)	OR (95% CI)	*p*	Overall *p*	OR (95% CI)	*p*	Overall *p*
I walk or perform equivalent physical activity for at least 1 h/day in my daily life								
Yes	1586 (48.0)	204 (34.6)	0.60 (0.50–0.73)	<0.001		0.69 (0.57–0.83)	<0.001	
No	1721 (52.0)	386 (65.4)	reference			reference		
I walk faster than people of the same age and sex								
Yes	1290 (39.0)	189 (32.0)	0.76 (0.63–0.91)	0.004		0.96 (0.78–1.17)	0.665	
No	2017 (61.0)	401 (68.0)	reference			reference		
Chewing situation								
Able to chew anything	2352 (71.1)	359 (60.8)	reference			reference		
Difficulty in chewing	955 (28.9)	231 (39.2)	1.56 (1.30–1.87)	<0.001		1.40 (1.15–1.69)	0.001	
I eat faster than other people								
Fast	414 (12.5)	67 (11.4)	1.06 (0.80–1.40)	0.706	0.001	1.07 (0.80–1.42)	0.671	0.149
Normal	2268 (68.6)	360 (61.0)	reference			reference		
Slow	625 (18.9)	163 (27.6)	1.55 (1.26–1.90)	<0.001		1.26 (1.011.57)	0.039	
I skip breakfast three or more times a week								
Yes	155 (4.7)	33 (5.6)	reference			reference		
No	3152 (95.3)	557 (94.4)	0.86 (0.58–1.27)	0.458		1.01 (0.67–1.51)	0.972	
Smoking habits								
Smoke	3221 (97.4)	566 (95.9)	reference			reference		
Do not smoke	86 (2.6)	24 (4.1)	1.58 (0.99–2.53)	0.057		1.47 (0.91–2.40)	0.118	
Drinking habits								
Drink daily	458 (13.8)	78 (13.2)	0.86 (0.65–1.15)	0.303	0.142	0.91 (0.68–1.22)	0.541	0.419
Drink occasionally	476 (14.4)	68 (11.5)	0.78 (0.59–1.03)	0.075		0.89 (0.67–1.19)	0.434	
Rarely drink	2373 (71.8)	444 (75.3)	reference			reference		
I get enough rest from sleep								
Yes	2763 (83.6)	472 (80.0)	reference			reference		
No	544 (16.4)	118 (20.0)	1.31 (1.05–1.64)	0.017		1.18 (0.93–1.48	0.168	
Weight loss in the previous year								
≤5%	2928 (88.5)	485 (82.2)	reference		<0.001	reference		<0.001
5–10%	327 (9.9)	73 (12.4)	1.32 (1.00–1.73)	0.048		1.19 (0.90–1.58)	0.223	
>10%	52 (1.6)	32 (5.4)	3.66 (2.31–5.78)	<0.001		2.74 (1.69–4.43)	<0.001	
BMI categories								
Underweight	403 (12.2)	108 (18.3)	1.70 (1.32–2.17)	<0.001	0.001	1.38 (1.06–1.80)	0.017	0.057
Slightly underweight	470 (14.2)	91 (15.4)	1.20 (0.92–1.55)	0.175		1.12 (0.86–1.46)	0.393	
Normal	1899 (57.4)	299 (50.7)	reference			reference		
Overweight	535 (16.2)	92 (15.6)	1.12 (0.87–1.45)	0.374		1.08 (0.83–1.40)	0.583	

Data are expressed as odds ratios (OR) and 95% confidence intervals (CIs) upon multiple logistic regression analysis. ^†^ Adjusted for sex, age, metabolic diseases, musculoskeletal diseases, cardiovascular diseases, kidney diseases, respiratory inflammatory diseases, malignant neoplasms, walking at least 1 h/day, walking speed, chewing situation, eating speed, skipping breakfast, smoking habits, drinking habits, getting enough rest from sleep, weight loss in the previous year, BMI categories.

## Data Availability

These data are not appropriate for public disclosure owing to ethical issues. If you are a researcher who is interested in an analysis using these data, please request access to the confidential data from the bioethics review committee at Nagoya University Graduate School of Medicine.

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
