# Peer review of "Factors Associated with Low Albumin in Community-Dwelling Older Adults Aged 75 Years and Above"

_ijerph, 2023, doi:10.3390/ijerph20216994_

Round 1
Reviewer 1 Report
Comments and Suggestions for Authors
I really appreciate the way you presented the obtained data. The research methodology is presented in a very comprehensive way, which is a positive assessment of this text. Similarly, the results are presented very clearly and legibly.
The introduction is the weakest part of the entire text. There are no theoretical references to the literature. Currently, this introduction has a bit of a discussion nature. This part, and therefore the bibliography, should be expanded.
I suggest that at the beginning of the introduction it is better to describe the albumins themselves. Due to the interdisciplinary nature of the journal, it is used by scientists who do not have detailed medical knowledge (we are talking about public health, which is dealt with by researchers from various sciences). A better description of the studied proteins will allow readers to have a more complete understanding of this text.
At this point, the introduction is somewhat of a discussion - the authors refer to research by other scientists. This is, of course, of great value, but I lack a "theoretical" basis here.
Author Response
Dear Reviewer:
We greatly thank referees for careful reading our manuscript and for giving us useful comments.
According to the comments, we have revised the manuscript "Factors associated with low albumin in community-dwelling older adults aged 75 years and above" (ID ijerph-2658135).
Revisions in the text have been shown using yellow highlight for main additions.
Comments and Suggestions for Authors
I really appreciate the way you presented the obtained data. The research methodology is presented in a very comprehensive way, which is a positive assessment of this text. Similarly, the results are presented very clearly and legibly.
The introduction is the weakest part of the entire text. There are no theoretical references to the literature. Currently, this introduction has a bit of a discussion nature. This part, and therefore the bibliography, should be expanded.
I suggest that at the beginning of the introduction it is better to describe the albumins themselves. Due to the interdisciplinary nature of the journal, it is used by scientists who do not have detailed medical knowledge (we are talking about public health, which is dealt with by researchers from various sciences). A better description of the studied proteins will allow readers to have a more complete understanding of this text.
At this point, the introduction is somewhat of a discussion - the authors refer to research by other scientists. This is, of course, of great value, but I lack a "theoretical" basis here.
Response:
According to the comment, we have revised the introduction. At the beginning of the introduction, it has described the albumins themselves. Additionally, introduction has revised to be more theoretical and easier to understand. (page 1, line 28 to page 2, line 56)
Reviewer 2 Report
Comments and Suggestions for Authors
Thank you for the opportunity to review this article. The aim of the authors was to investigate the relationship between serum albumin and several frailty-related parameters. First of all, I want to congratulate the authors: the article is well-written, the scope of the research is clear, and it is easy to follow the rationale.
I have some comments:
- Page 2, line 92: I believe that paragraph “2.2 Survey content” is presented too briefly. I encourage the authors to provide more detail.
- Page 3, line 112: In the same paragraph, regarding the threshold level considered clinically critical, the authors report it to be 3.5 g/dL, bringing some literature references to support it. Subsequently, they state that for this study the threshold value used was 3.8 g/dL. I invite them to explain why they made this choice, which is not in accordance with the literature presented.
- Page 3, lines 119-121: Two questions in the proposed questionnaire state "I walk faster than people of the same age and sex," and "I eat faster than other people". Without objective elements that can confirm what is the feelings of the subjects, I wonder how the answers to these questions can be considered trustworthy. Given that walking speed is a relevant element in the article's conclusions, I urge the authors to address this issue.
- Page 6, lines 206-208: there is the repetition of the phrase "had significantly low odds of low albumin levels".
- Page 8, lines 233-235: the authors state that "older adults who walk faster than their peers are expected to be quite active and maintain their muscle strength and physical function." I invite them to include a bibliographic entry, supporting this statement.
Since the article surprisingly concludes that walking at least 1 hour/day and walking fast is associated with low albumin levels, and this is contradictory to WHO guidelines (as reported by the authors), I urge the authors to focus on this. I suggest that they give their own interpretation of this unexpected result, or refer to it in the description of the study's limitations if they consider that there may be some bias in the patients' completion of the questionnaire.
Author Response
Dear Reviewer:
We greatly thank referees for careful reading our manuscript and for giving us useful comments.
According to the comments, we have revised the manuscript "Factors associated with low albumin in community-dwelling older adults aged 75 years and above" (ID ijerph-2658135).
Revisions in the text have been shown using yellow highlight for main additions.
Comments and Suggestions for Authors
Page 2, line 92: I believe that paragraph “2.2 Survey content” is presented too briefly. I encourage the authors to provide more detail.
Response:
According to the comment, we have added information about the survey content; “In Japan, the health check-up for the elderly aged 75 and over is conducted under the jurisdiction of the Ministry of Health, Labour, and Welfare using a standardized protocol. Therefore, the physical measurement items, blood test items, and lifestyle questionnaires are nearly uniform in the health checkup of all 42 municipalities in Gifu Prefecture covered in this study.” (page 3, line 99 to line 103)
Page 3, line 112: In the same paragraph, regarding the threshold level considered clinically critical, the authors report it to be 3.5 g/dL, bringing some literature references to support it. Subsequently, they state that for this study the threshold value used was 3.8 g/dL. I invite them to explain why they made this choice, which is not in accordance with the literature presented.
Response:
A cut-off value of 3.5 g/dL is not used in this study, therefore we deleted description of A cut-off of 3.5 g/dL. We have revised: “Previous studies on the general population have demonstrated that the inverse relationship between serum albumin and risk of death, hospitalization, and frailty are present [7,8,10]. A cut-off value of 3.8 g/dL on albumin has been used among community-dwelling older adults, and has been shown to have a high risk of sarcopenia and mortality risk [18-20].” (page 3, line 118 to line 122)
Page 3, lines 119-121: Two questions in the proposed questionnaire state "I walk faster than people of the same age and sex," and "I eat faster than other people". Without objective elements that can confirm what is the feelings of the subjects, I wonder how the answers to these questions can be considered trustworthy. Given that walking speed is a relevant element in the article's conclusions, I urge the authors to address this issue.
Response:
The questions used in this study were those that are standard in Japan for medical examinations of the elderly aged 75 years and older. As the reviewer mentioned, responses to the questionnaire in this study were self-reported and included subjective. Thus, we have added the limitation; “Responses to the questionnaire in this study were self-reported and included subjective statements such as " I walk faster than people of the same age and sex." Previous study has demonstrated that subjectively faster walking speeds are associated with a lower risk of heart failure and a variety of cardiovascular disease events in the general population, and have reported the utility of subjective walking speed in primary prevention [38]. However, it has limitations in terms of accuracy of responses.” (page 9, line 327 to line 332)
Page 6, lines 206-208: there is the repetition of the phrase "had significantly low odds of low albumin levels".
Response:
According to the comments, we have revised: “had significantly lower odds of low albumin levels.” (page 6, line 216 to line 217).
Page 8, lines 233-235: the authors state that "older adults who walk faster than their peers are expected to be quite active and maintain their muscle strength and physical function." I invite them to include a bibliographic entry, supporting this statement.
Response:
According to the comment, we had added the references. (page 8, line 246)
Since the article surprisingly concludes that walking at least 1 hour/day and walking fast is associated with low albumin levels, and this is contradictory to WHO guidelines (as reported by the authors), I urge the authors to focus on this. I suggest that they give their own interpretation of this unexpected result, or refer to it in the description of the study's limitations if they consider that there may be some bias in the patients' completion of the questionnaire.
Response:
We do not consider that our results have been contradictory to WHO guidelines. The questions in this study were standard in Japan for medical examinations, and not based on WHO guidelines. Nevertheless, we consider that older adults walking for at least 1 hour/day are older adults who sufficiently achieve the WHO guideline recommendations for physical activity (but only for time). This result has been not unexpected.
Reviewer 3 Report
Comments and Suggestions for Authors
Very clear and well written manuscript, focusing a relevant issue and adding interesting information to current knowledge. I consider that only minor issues need to be addressed.
1. When referring the association of "walking at least 1 h/day" with albumin, please indicate the direction of such relationship. Currently, some paragraphs (e.g. abstract and 1st paragraph of the discussion) may be interpreted as if low albumin is indeed associated with walking > 1 hour/day. Please refer to the category (NOT walking...) instead of the variable.
2. Please add a reference for 3.8 g/dL as cutpoint for albumin and justify its use, specially considering the focus on the 3.5 cutoff in lines 109-112.
3. Despite the large sample size, the limitations section should be improved regarding potential differences between the overal 327,498 older adults, the 70,189 with checkups, and the 27,303 with no missing data.
4. Please refer explicitly if standard procedures are used for data collection in the checkups, as this may be very different between countries.
5. I suggest using "weight loss in previous year" instead of "weight loss rate" (text and tables), for easiness of understanding.
6. Please clarify why/where X2 vs. Fisher's tests were used.
7. Also, why was age divided in categories (instead of analysed directly) and then treated as nominal (by using X2/Fisher's test) instead of ordinal, as it really is?
8. Line 169: correct "Almost diseases" to "Almost all diseases".
9. In tables 1 and 2, the lines referring to not having a certain disease or condition may be deleted. This will inprove their readability.
10. In table 2, for independent variables with more than 2 categories, please provide the overall p value (not only the p value for the comparison of each category with the reference one).
Author Response
Dear Reviewer:
We greatly thank referees for careful reading our manuscript and for giving us useful comments.
According to the comments, we have revised the manuscript "Factors associated with low albumin in community-dwelling older adults aged 75 years and above" (ID ijerph-2658135).
Revisions in the text have been shown using yellow highlight for main additions.
Comments and Suggestions for Authors
1. When referring the association of "walking at least 1 h/day" with albumin, please indicate the direction of such relationship. Currently, some paragraphs (e.g. abstract and 1st paragraph of the discussion) may be interpreted as if low albumin is indeed associated with walking > 1 hour/day. Please refer to the category (NOT walking...) instead of the variable.
Response:
According to the comments, we have revised: (ex.) “low albumin levels (<3.8 g/dL) were associated with smoking, not walking at least 1 hour/day, not fast walking speed, difficulty in chewing, slow eating speed, weight loss in previous year, and underweight BMI classification (<18.5 kg/m2).” (page 7, line 232 to line 238)
2. Please add a reference for 3.8 g/dL as cutpoint for albumin and justify its use, specially considering the focus on the 3.5 cutoff in lines 109-112.
Response:
A cut-off value of 3.5 g/dL is not used in this study, therefore we deleted description of A cut-off of 3.5 g/dL. We have revised: “Previous studies on the general population have demonstrated that the inverse relationship between serum albumin and risk of death, hospitalization, and frailty are present [7,8,10]. A cut-off value of 3.8 g/dL on albumin has been used among community-dwelling older adults, and has been shown to have a high risk of sarcopenia and mortality risk [18-20].” (page 3, line 118 to line 122)
3. Despite the large sample size, the limitations section should be improved regarding potential differences between the overal 327,498 older adults, the 70,189 with checkups, and the 27,303 with no missing data.
Response:
According to the comment, we have revised the limitations section, and added “In this study, among overall 327,498 older adults, there were 70,189 with health checkup, and the 27,303 with no missing data. Therefore, there is a possibility that the data collect-ed was from relatively healthy individuals.” (page 9, line 322 to line 324)
4. Please refer explicitly if standard procedures are used for data collection in the checkups, as this may be very different between countries.
Response:
According to the comment, we have added information about the database system used in this study: “The Kokuho Database includes the data such as the health checkups, medical, and nursing care, from the National Health Insurance for individuals aged <75 years and the Latter-stage Elderly Medical Care System for individuals aged ≥75 years among residents in each prefecture. The health checkup for the elderly aged 75 and over is conducted under the jurisdiction of the Ministry of Health, Labour, and Welfare using a standardized protocol, and the data is registered in this system.” (page 2, line 80 to line 86)
5. I suggest using "weight loss in previous year" instead of "weight loss rate" (text and tables), for easiness of understanding.
Response:
According to the comment, we have corrected to “weight loss in previous year.”
6.Please clarify why/where X2 vs. Fisher's tests were used.
7.Also, why was age divided in categories (instead of analysed directly) and then treated as nominal (by using X2/Fisher's test) instead of ordinal, as it really is?
Response:
According to the comment, we have revised statistical analysis and Table 1. Also, age divided in categories is not necessary for overall goals of the study. Therefore, we deleted this category.
8. Line 169: correct "Almost diseases" to "Almost all diseases".
Response:
According to the comment, we have corrected to “Almost all diseases.” (page 4, line 179)
9. In tables 1 and 2, the lines referring to not having a certain disease or condition may be deleted. This will improve their readability.
Response:
According to the comment, we have deleted the lines referring to not having a certain disease or condition in Table 1. In Table 2, we have not deleted them to show the data in the reference.
10. In table 2, for independent variables with more than 2 categories, please provide the overall p value (not only the p value for the comparison of each category with the reference one).
Response:
According to the comment, we had added overall p-value for categorical variable in logistic regression analysis in Table 2 and Table 3.
Round 2
Reviewer 2 Report
Comments and Suggestions for Authors
The authors responded to my concerns and revised the paper in accordance with my suggestions.
I am in favor of publishing the article in the present form.